# Uracil in the carbonaceous asteroid (162173) Ryugu

Yasuhiro Oba [1] ✉, Toshiki Koga [2], Yoshinori Takano [2,3] ✉, Nanako O. Ogawa [2], Naohiko Ohkouchi[2], Kazunori Sasaki[3,4], Hajime Sato[4], Daniel P. Glavin [5], Jason P. Dworkin [5], Hiroshi Naraoka [6], Shogo Tachibana [7,8], Hisayoshi Yurimoto [9], Tomoki Nakamura[10], Takaaki Noguchi [11], Ryuji Okazaki[6], Hikaru Yabuta [12], Kanako Sakamoto [8], Toru Yada [8], Masahiro Nishimura[8], Aiko Nakato[8], Akiko Miyazaki [8], Kasumi Yogata [8], Masanao Abe[8], Tatsuaki Okada [8], Tomohiro Usui [8], Makoto Yoshikawa[8], Takanao Saiki[8], Satoshi Tanaka[8], Fuyuto Terui[13], Satoru Nakazawa [8], Sei-ichiro Watanabe [14], Yuichi Tsuda[8] & Hayabusa2-initial-analysis SOM team*

The pristine sample from the near-Earth carbonaceous asteroid (162173) Ryugu collected by the Hayabusa2 spacecraft enabled us to analyze the pristine extraterrestrial material without uncontrolled exposure to the Earth's atmosphere and biosphere. The initial analysis team for the soluble organic matter reported the detection of wide variety of organic molecules including racemic amino acids in the Ryugu samples. Here we report the detection of uracil, one of the four nucleobases in ribonucleic acid, in aqueous extracts from Ryugu samples. In addition, nicotinic acid (niacin, a $B_3$ vitamer), its derivatives, and imidazoles were detected in search for nitrogen heterocyclic molecules. The observed difference in the concentration of uracil between A0106 and C0107 may be related to the possible differences in the degree of alteration induced by energetic particles such as ultraviolet photons and cosmic rays. The present study strongly suggests that such molecules of prebiotic interest commonly formed in carbonaceous asteroids including Ryugu and were delivered to the early Earth.

Hayabusa2 spacecraft successfully delivered total 5.4 g of pristine samples, collected during two touchdown operations, from the C-type near-Earth asteroid (162173) Ryugu on 6 December 2020[1]. Initial analyses of the samples revealed the similarity with Ivuna-type (CI) carbonaceous chondrites based on spectroscopy, mineralogy, and elemental/isotopic compositions[2–5]. Surveys of soluble organic matter (SOM) in the Ryugu samples have also been performed in the initial analysis, revealing that a wide range of organic compounds such as racemic non-protein type amino acids, alkylamines, carboxylic acids, polycyclic aromatic hydrocarbons (PAHs), and nitrogen heterocyclic molecules are present in the aqueous or organic extracts[3,6]. Then, if

Ryugu had a common parent body with CI chondrites, we would expect that other classes of organic compounds, in particular nucleobases, to be present in the Ryugu samples, as has been demonstrated in previous studies on the detection of uracil in the Orgueil CI meteorite[7–10].

We have recently developed an analytical method for the small-scale detection and identification of nucleobases at parts per billion (ppb) to parts per trillion (ppt) levels using high-performance liquid chromatography coupled with electrospray ionization high-resolution mass spectrometry (HPLC/ESI-HRMS)[11,12]. In the aqueous extracts from the Murchison CM meteorite, for which intensive study of organics

have been made since its fall in 1969, we successfully detected all five canonical nucleobases (adenine, guanine, cytosine, thymine, uracil) at the concentrations ranging from 4 to 72 ppb[11]. Such highly sensitive methods are well suited for Ryugu analyses where the available sample mass is limited.

Here, we search for nucleobases and other classes of nitrogen (N)-heterocyclic molecules (Supplementary Fig. 1), as well as their alkylated homologs, indigenous to the Ryugu samples A0106 and C0107 (Fig. 1), which were collected from the first and second touchdown sites of the asteroid Ryugu. Such N-heterocyclic molecules including nucleobases would be a good target for better understanding chemical evolution as they are considered to play an important role both as reactants and catalysts for the synthesis of further complex organic molecules such as peptide nucleic acids and nucleotides in prebiotic evolution[13–15] (Supplementary Fig. 2). The Ryugu samples (~10 mg) are subjected to hot-water extraction at 105 °C for 20 hours, followed by acid hydrolysis (see "Methods" section). As the most suitable reference samples of Ryugu, we also analyze the hot water extract from the Orgueil meteorite and its acid-hydrolysate to compare the nucleobase contents between the Ryugu and Orgueil meteorite.

## Results and discussion

### Bulk properties of the Ryugu samples

The bulk elemental and isotopic analyses of both samples showed very good similarities between Ryugu and CI chondrites[5,6] (Fig. 2 and Supplementary Table 1). The infrared reflectance spectral analysis of those Ryugu samples indicated that the signals derived from hydrous silicate minerals (i.e., 2.72 μm for OH absorption) and organic matter (i.e., 3.1 μm and 3.4 μm for CH and NH absorption bands, respectively) were mostly homogeneous between A0106 and C0107 (Fig. 3) and had similar characteristics to a profile of CI chondritic features[2].

### Detection of uracil

Figure 4 shows mass chromatograms at the mass-to-charge ratio ($m/z$) of 113.0346 Da (atomic mass unit) corresponding to that of protonated uracil ($C_4H_4N_2O_2 + H^+$) for the acid hydrolysates of the A0106 and C0107 extracts. Several peaks were observed above the background level in the sample chromatograms. Since there are no peaks in the mass chromatograms for the serpentine blank at the corresponding

$m/z$, the observed peaks must be indigenous to the samples. Based on the comparison with the retention times using the authentic standard reagents of uracil and its structural isomers (2-imidazole-carboxylic acid and 4-imidazole-carboxylic acid; hereafter 2-ICA and 4-ICA, respectively; Supplementary Fig. 1), the observed peak at ~16 min for the Ryugu samples is certainly derived from uracil. This peak was also observed in the extract from the Orgueil meteorite (Fig. 4) and the MS/MS analysis (see Methods) for the peak at ~16 min showed a clear match on the mass fragmentation pattern with that for the authentic standard of uracil (Supplementary Fig. 3). The identification of uracil in the Orgueil extracts further strengthens our conclusion that the detected peak at 16 min in the Ryugu extracts is surely derived from uracil. The simultaneous detection of uracil from the A0106 and C0107 extracts by a different separation technique using a capillary electrophoresis-high resolution mass spectrometry (CE-HRMS) instrument is important evidence that reliably supports the analytical detection of uracil with HPLC/ESI-HRMS as discussed above (Fig. 5).

The concentration of uracil was 11 ± 6 and 32 ± 9 ppb in the A0106 and C0107 samples, respectively, which were lower than that detected in the same extract from the Orgueil meteorite analyzed in the present study (140 ppb, Table 1), and lower than that reported previously (73 ppb)[8], but comparable to that in CM2 chondrites (Table 1). It is likely that acid hydrolysis may have liberated or synthesized uracil from its derivatives or precursor(s) in the hot water extracts, as the case for amino acids in carbonaceous meteorites[16]. In fact, in the Orgueil extract, the concentration of uracil increased by a factor 1.5 after acid hydrolysis (Table 1). Applying this factor to the Ryugu samples, the concentration of uracil before acid hydrolysis is estimated to be 7 ± 4 and 21 ± 6 ppb in the A0106 and C0107 samples, respectively. This implies that uracil has been present in the Ryugu samples as a labile form to some extent, which is well consistent with the detection of uracil in the hot water extract from both Ryugu samples before acid hydrolysis (Fig. 5). Other DNA/RNA nucleobases were not positively identified in both the A0106 and C0107 samples. This does not necessarily exclude a possibility that those nucleobases are present in the Ryugu samples, but just they may be below the detection limit under the experimental conditions employed (see Supplementary Information for the non-detection of purine nucleobases). Note that any cytosine which was present may have been deaminated to yield

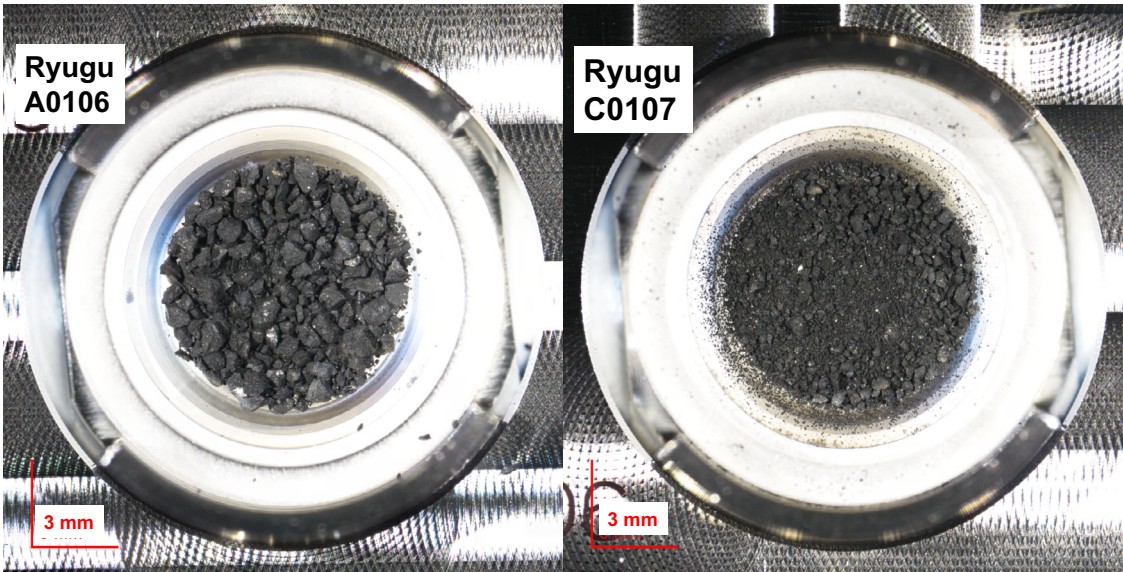

**Fig. 1 | Sample returned from asteroid Ryugu (162173).** Photographs of initial samples A0106 (total 38.4 mg)[6] and C0107 (total 37.5 mg) from the asteroid Ryugu (162173) during the 1st touchdown sampling and 2nd touchdown sampling, respectively[1,2]. The photos were taken in the clean chamber of the curation facility at Japan Aerospace Exploration Agency before the sample distribution. The scale bar represents 3 mm (red line).

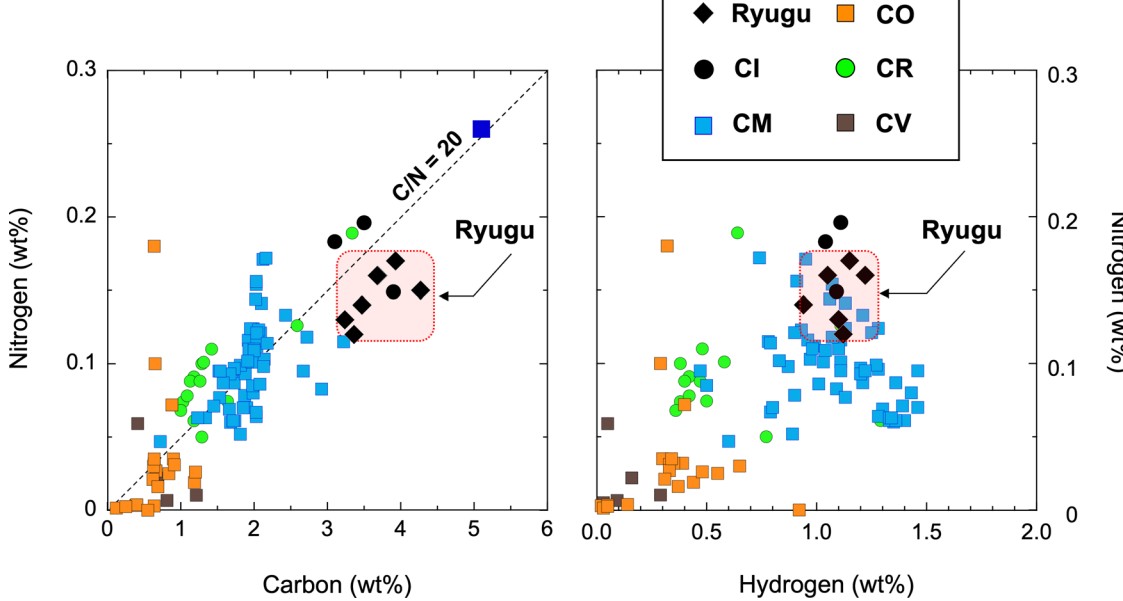

**Fig. 2 | Elemental compositions of the Ryugu samples.** The elemental profiles of A0106 (i.e., 1st touchdown site) and C0107 (i.e., 2nd touchdown site) for carbon (C, wt%), nitrogen (N, wt%), and hydrogen (H, wt%). with those of reference carbonaceous chondrites (CI, CM, CO, CR, and CV; Naraoka et al.[6] and the supplementary information). Based on repeated measurements of very fine-grained samples, both A0106 and C0107 were found to have similar elemental abundances (C, N, H). The blue square stands for bulk C (5.1 ± 0.4 wt%) and N (0.26 ± 0.01 wt%) profiles of the Zag clast[53].

uracil upon hot water extraction for 20 h at 105 °C, followed by acid hydrolysis for the same duration at 105 °C, as has been demonstrated in laboratory experiments[17,18]. Hence, the possibility of cytosine in the Ryugu samples prior to workup cannot be excluded. The upper limit for the concentrations of uracil precursors including cytosine can be estimated from the difference in the concentration of uracil before and after acid hydrolysis, to be 4 and 11 ppb in the A0106 and C0107 samples, respectively. There could be other formation mechanisms of uracil in the Ryugu sample other than hydrolysis of its chemical precursors. Since Ryugu appears to not have heated above 100 °C after aqueous alteration[5] and above 200 °C by radiative heating at the -1 m depth below the surface after its orbital transition to near-Earth orbit[19], formation mechanisms at relatively low temperatures may play a role for their formation. Photochemical reactions of interstellar ices which are composed of $H_2O$, $NH_3$, $CH_3OH$, and other simple molecules, followed by accretion into the asteroid upon the formation of the Solar system, may be one of the possible pathways for the synthesis of nucleobases[12,20]. Nucleobases including uracil could also be synthesized through the side-addition to pyrimidine via photochemical reactions in ices[21] and via energetic processes to formamide on asteroids[22].

## Detection of other N-heterocycles

In addition to uracil, its structural isomers 2-ICA and 4-ICA were also identified in the chromatogram at the m/z of 113.0346 (Fig. 4). The presence of organics with carboxyl groups is consistent with the finding of low molecular weight organic acids in the previous report[6]. Imidazoles have long been considered to act as a catalyst on the activation of organic monomers like nucleotides and amino acids[13,14,23], thus they are expected to play an important role on the chemical evolution on asteroids and also on the early Earth. The concentration of 4-ICA is in the same range of uracil, while that of 2-ICA is clearly lower than other two isomers (Table 1). The relatively low abundance of 2-ICA is well consistent with that observed in the Orgueil meteorite (Table 1) and CM2 carbonaceous meteorites[11]. Note that imidazoles which substituted at the 4-position of the ring are thought to form from hydrogen cyanide (HCN) via a proposed prebiotic pathway

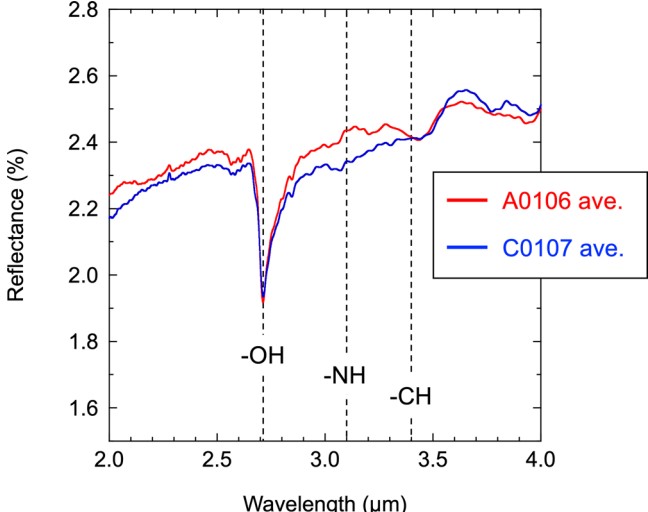

**Fig. 3 | Infrared spectra of the Ryugu samples.** The raw profiles of infrared reflectance spectra for Ryugu initial bulk samples (A0106 and C0107) within 2.0–4.0 μm. The data acquisition of this non-destructive analysis is described in the literature[2]. Signals for important functional groups (OH, NH, CH) are indicated in each band by dotted lines. The data management policy is declared in the "Data availability" section of this report.

toward the formation of purine nucleobases[24,25]. If this is the case on carbonaceous asteroids, the preference of 4-ICA over 2-ICA in the Ryugu samples and meteorites may demonstrate such a molecular evolution pathway from HCN to purine nucleobases in a natural environment.

We also detected pyridine-based species in the acid hydrolysate of Ryugu samples. Figure 6 shows mass chromatograms at the m/z of 124.0393, which corresponds to the protonated ion of nicotinic acid ($C_6H_5NO_2$). Nicotinic acid, known as niacin a $B_3$ vitamer, and its structural isomer isonicotinic acid were identified in the A0106 and

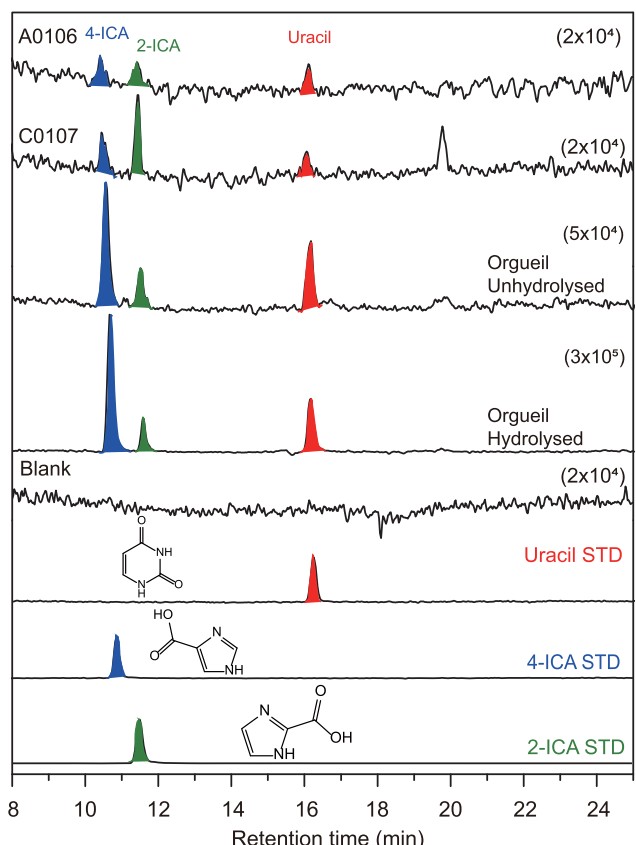

**Fig. 4 | Uracil ($C_4H_4N_2O_2$) with their structural isomers detected in Ryugu A0106 and C0107 samples and Orgueil meteorites.** Mass chromatograms at the $m/z$ of 113.0346 which corresponds to the protonated ion of exact mass number for uracil in the acid hydrolysate of the hot water extracts from the A0106 and C0107 samples. For Orgueil, mass chromatograms measured before and after acid hydrolysis were also shown. Data for the blank sample and authentic standards are also shown for comparison. Numbers in parenthesis represents the range for the vertical-axis in the sample chromatograms.

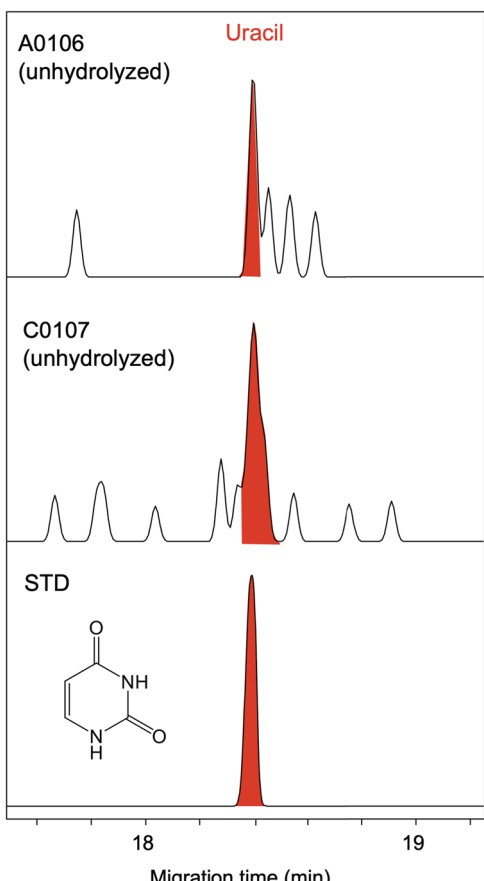

**Fig. 5 | Uracil in the hot water extracts from the Ryugu samples.** Cross-validation and high-resolution molecular identification of uracil obtained from hot water extracts (unhydrolyzed fraction) for Ryugu A0106 and C0107 by using capillary electrophoresis (CE) coupled with high resolution orbitrap mass spectrometry (CE-HRMS). Data for the authentic standard is also shown for comparison. The signal of red color represents uracil on the migration time (18.4 min).

C0107 samples (Table 1). These molecules were also identified in the hot water extract and its acid-hydrolysate from the Orgueil meteorite (Fig. 6), although their concentrations are one order of magnitude larger than those in the Ryugu samples. These molecules have also been identified in several carbonaceous chondrites and are more abundant than nucleobases[11,26]. Their structural isomer, picolinic acid (Supplementary Fig. 1), was identified neither in Ryugu nor in carbonaceous chondrites (Fig. 6). This is well consistent with the distribution of laboratory products which were formed by photochemical reactions of interstellar ice analogs containing $H_2O$, CO, $CH_3OH$, and $NH_3$ at 10 K[11,12] (Supplementary Fig. 4), also suggesting a contribution from low-temperature photochemical reactions to the origin of N-heterocycles in the Ryugu samples. Nicotinamide ($C_6H_6N_2O$), another $B_3$ vitamer, was not detected in the acid-hydrolysates from the A0106 and C0107 samples. However, this molecule may have been originally present in the Ryugu samples, but would have decomposed upon acid hydrolysis of the hot water extract. In fact, nicotinamide and its structural isomer isonicotinamide were detected in the hot water extract from the Orgueil meteorite before acid hydrolysis, while they were not after the acid hydrolysis (Supplementary Fig. 5 and Table 1), which supports our assumption above. Since nicotiniamide and isonicotinamide are also present in CM2 Murchison meteorites[11], they may also be common in CI type carbonaceous asteroids.

Alkylated homologs of uracil, as well as those of other N-heterocyclic molecules such as pyrimidine and nicotinic acid with

the number of carbon atoms up to 30 were detected in the methanol extracts from both A0106 and C0107 samples (Fig. 7 and Supplementary Figs. 6 and 7). The presence of the alkylated homologs of uracil coupled with the careful contamination control and chain of custody of the Hayabusa2 sample[27–29] strongly suggests that they are not formed under the control of terrestrial biology but formed through a series of abiotic processes in extraterrestrial environments. Alkylated homologs of N-heterocyclic molecules produced by photochemical reactions of interstellar ice analogs have shorter alkylchains than that observed in the Ryugu samples and meteorites[11], which strongly suggests that processes in asteroidal environments contributed to elongate alkylchains in those classes of molecules.

### Comparison of N-heterocycles contents between A0106 and C0107

Variations in the concentrations of N-heterocyclic molecules including uracil between A0106 and C0107 samples may be related to the possible differences in the degree of alteration processes induced by energetic particles such as ultraviolet photons and cosmic rays since material in Chamber C may have been ~1 m below the surface of Ryugu for several million years[3,30]. Organic molecules in the surface materials would have experienced energetic processes more extensively than those in the subsurface materials, which potentially causes preferential degradation of molecules at the surface[31–33] (Fig. 8). In fact, laboratory experiments have demonstrated that uracil is not strong against energetic processes such as

**Table 1 | Qualitative and quantitative summary (in ppb) of uracil and other N-heterocyclic molecules identified in the Ryugu samples (A0106, C0107) and Orgueil meteorite**

| | A0106 | C0107 | Orgueil | | Orgueil[c] | Murchison[d] | Murray[d] | Tagish Lake[d] |
|---|---|---|---|---|---|---|---|---|
| | Hydrolyzed | Hydrolyzed | Unhydrolyzed | Hydrolyzed | H₂O and HCOOH extracts | H₂O extracts | H₂O extracts | H₂O extracts |
| Uracil | 11 ± 6[a] | 32 ± 9 | 92 | 140 | 73 | 15 | 37 | 4 |
| Imidazole-2-carboxylic acid | 6 | 9 ±1 | 12 | 12 | – | 19 | 14 | 4 |
| Imidazole-4-carboxylic acid | 17 ± 3 | 19 ± 3 | 136 | 218 | – | 90 | 99 | 19 |
| Nicotinic acid | 49 ±1 | 99 ± 4 | 602 | 715 | – | 91 | 626 | 108 |
| Picolinic acid | n.d.[b] | n.d. | n.d. | n.d. | – | – | – | – |
| Isonicotinic acid | 49 ± 20 | 62 ± 23 | 237 | 203 | – | 53 | 307 | 118 |
| Nicotinamide | n.d. | n.d. | 214 | n.d. | – | 10 | – | – |
| Isonicotinamide | n.d. | n.d. | 38 | n.d. | – | 3 | – | – |
| Picolinamide | n.d. | n.d. | n.d. | n.d. | – | 1 | – | – |

[a]Averaged value with an error on two measurements.
[b]Not detected.
[c]Stoks and Schwartz[8].
[d]Oba et al.[11].

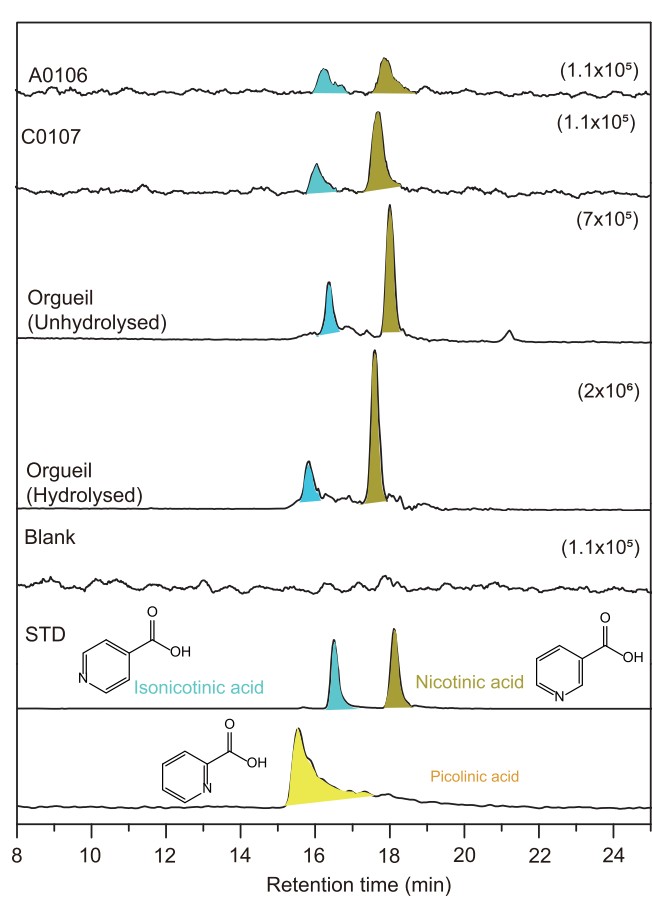

**Fig. 6 | nicotinic acid (C₆H₅NO₂) with their structural isomers detected in Ryugu A0106 and C0107 samples and Orgueil meteorites.** Mass chromatograms at the *m/z* of 124.0393 which corresponds to the protonated ion of exact mass number for nicotinic acid in the acid hydrolysate of the hot water extracts from the A0106 and C0107 samples. For Orgueil, mass chromatograms measured before and after acid hydrolysis were also shown. Data for the blank sample and authentic standards are also shown for comparison. Numbers in parenthesis represents the range for the vertical-axis in the sample chromatograms.

N-heterocycles such as nicotinic acids and other five- and six-membered N-heterocyclic molecules (Table 1). In contrast, the opposite is true for some polycyclic aromatic hydrocarbons (PAHs) such as naphthalene and pyrene where these PAHs in the A0106 are more abundant than those in the C0107[37]. The observed difference may be indicative of different histories which PAHs and N-heterocycles have experienced on the asteroid Ryugu. For example, PAHs could be produced as a result of surface degradation process of precursor molecules by photolysis and/or radiolysis, while the formation of N-heterocyclic molecules may be less effective compared to their degradation through the same processes. Further studies are necessary to elucidate variations in the concentrations of molecules between surface and subsurface materials of the Ryugu.

**Possible formation mechanisms of N-heterocycles and future perspective**

Naraoka et al.[38] proposed that aldol condensation and formose reaction, both of which requires aldehydes such as formaldehyde, with ammonia are a formation pathway of N-heterocyclic molecules in carbonaceous asteroids. The source of ammonia and formaldehyde in the asteroid Ryugu would be the key factor to constrain the validity of this assumption. In addition, HCN may also be used for the synthesis of nucleobases as explained above[24,25]. So far, these three molecules have not been identified in the Ryugu samples. However, it is highly likely that they were present in the asteroid Ryugu if it is composed of comet-like materials[3,39] since HCN, ammonia, and formaldehyde have been identified in cometary ices[40] and cyanides have been detected in carbonaceous chondrites[41]. In fact, the N–H feature has been observed in the Ryugu samples[4,42], which supports the assumption above. As for the other candidates for the source of nitrogen, molecular nitrogen (N₂) may also play a role for the synthesis of N-heterocycles regardless of its chemically inert nature[43] since it has been detected in cometary environments[44]. Hence, it is likely that availability of N-containing molecules such as ammonia, HCN, and/or their precursors should constrain the synthesis of N-heterocyclic molecules on the asteroid Ryugu. If hexamethylenetetramine was present in Ryugu, as the case for some carbonaceous meteorites including Murchison[45], it can play a role for the precursor of these molecules upon aqueous and/or thermal activities in Ryugu. Another possible candidate for the synthesis of uracil may be urea (NH₂CONH₂) as has been proposed previously[46]. In fact, urea is abundant in carbonaceous meteorites[47]. In addition, although urea was not detected in the present study, its presence in Ryugu was suggested in another research[3]. Urea can also

cosmic ray and ultraviolet photons[34,35]. While uracil in the subsurface (~5 cm) on planetary bodies can be protected by the surface minerals such as calcium carbonate, calcium sulfate, and kaolinite from radiolysis[36]. Similar variation has been observed for other

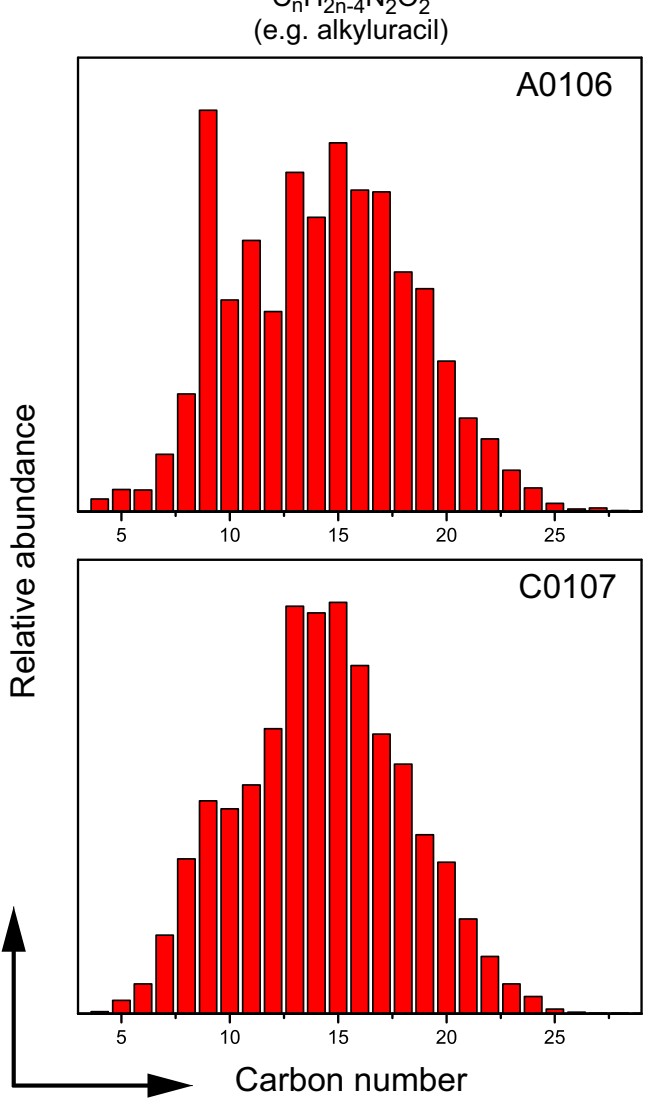

$C_nH_{2n-4}N_2O_2$
(e.g. alkyluracil)

A0106

C0107

Relative abundance

Carbon number

**Fig. 7 | Detection of CHNO molecules in the methanol extract from the Ryugu samples.** Relative peak area of alkylated analogs of CHNO molecules having a general formula of $C_nH_{2n-4}N_2O_2$ which include uracil and their structural isomers in the methanol extracts from the A0106 (top) and C0107 (bottom) samples with relevance to the number of carbon atoms ($n = 5$–$30$) in a molecule. The horizontal axis shows the number of carbon atoms and the vertical axis shows the sum of the peak area at each $m/z$ in arbitrary units.

be synthesized in photochemical reactions of representative components in interstellar ices at low temperatures[48,49].

Sample from near-Earth asteroid (101955) Bennu is scheduled in 2023 with subsequent detailed analysis in the NASA's Origins, Spectral Interpretation, Resource Identification and Security-Regolith Explorer (OSIRIS-REx)[50], further elucidation of the history and prebiotic chemistry inscribed in the CI-type asteroid Ryugu will be definitely important for understanding past and ongoing chemical evolution on both carbonaceous asteroids.

## Methods

### CNHSO contents and their isotopic compositions

We analyzed the elemental abundance of carbon (C, wt%), nitrogen (N, wt%), hydrogen (H, wt%), pyrolyzable oxygen (O, wt%), and sulfur (S, wt %) and the isotopic compositions of $\delta^{13}C$ (‰ vs. VPDB), $\delta^{15}N$ (‰ vs. Air), $\delta D$ (‰ vs. VSMOW), $\delta^{18}O$ (‰ vs. VSMOW) and $\delta^{34}S$ (‰ vs. VCDT), respectively[6]. For the total C, N, and S contents with their isotopic

compositions ($\delta^{13}C$, $\delta^{15}N$, $\delta^{34}S$), we used an ultrasensitive nano-EA/IRMS method (Flash EA1112 elemental analyzer/Conflo III interface/Delta Plus XP isotope-ratio mass spectrometer, Thermo Finnigan Co., Bremen)[6,51]. For the total H and pyrolyzable O with their isotopic compositions ($\delta D$, $\delta^{18}O$), we used a highly sensitive TC/EA/IRMS method (Delta Plus XL isotope-ratio mass spectrometer, Thermo Finnigan Co., Bremen)[52]. Prior to the Hayabusa2 samples, analytical validations using the nano-EA/IRMS system were performed during the rehearsal analyses and application studies of the carbonaceous chondrites[53,54].

### Extraction of organic molecules from samples

We extracted organic molecules from the A0106 (13.08 mg) and C0107 (10.73 mg) Ryugu samples, as well as 10.86 mg of Orgueil meteorite (CI type, from Denmark National Museum) with hot water at 105 °C for 20 hrs in a $N_2$-purged and flame-sealed glass ampoule. The water extract from the Ryugu samples was further hydrolyzed using ultra-pure 6 M HCl at 105 °C for 20 hrs of this extract ~20% was made available for this study. Since the majority was allocated acid-hydrolyzed for the detection of amino acids[6], nucleobases in the hot water extract were not available for analysis with HPLC/ESI-HRMS before acid hydrolysis. The use of the Orgueil extract was not so constrained, so nucleobases in the hot water extract were investigated before and after acid hydrolysis. The same procedure was repeated using baked (500 °C in air for 3 hr) serpentine powder (17.58 mg) as a procedural blank.

Separate aggregates of A0106 (17.15 mg) and C0107 (17.36 mg) were subjected to extractions by a series of organic solvents with an assist of ultrasonication (15 min) in the following order: hexane, dichloromethane, and methanol. After the sonication, the liquid extract was decanted after centrifugation (9677×$g$, 5 min). About 30% of the methanol extract was allocated for the detection of nucleobases and other N-heterocyclic molecules. For further details, please refer to Naraoka et al.[6].

### Analysis of nucleobases and N-heterocyclic molecules

The acid-hydrolysate of the hot water extracts from the Ryugu samples and the Orgueil meteorite was introduced into an HPLC/ESI-HRMS instrument with a mass resolution of 140,000 at a mass-to-charge ratio ($m/z$) of 200. The analytical system comprised an Ultimate 3000 and Q-Exactive Plus (Thermo Fischer Scientific) equipped with a reversed-phase separation column (HyperCarb™, 150 mm length × 1.0 mm i.d., particle size 5 μm, Thermo Fisher Scientific) at 40 °C and operated in positive ion mode. Validations of analytical methods were also performed using a 1290 Infinity II coupled with a 6230 time-of-flight mass spectrometer (Agilent Technologies). The eluent program for the HPLC was as follows: solvent A (water + 0.1% formic acid) and solvent B (acetonitrile + 0.1% formic acid) = 99:1 at t = 0 min, followed by a linear gradient of A:B = 70:30 at 20 min, and maintained at this ratio for 25 min. The flow rate was 50 μL/min and the column temperature was kept at 40 °C. The injection volume was 5 μL. The mass spectra were recorded in positive ESI mode with an $m/z$ range of 50–600 at the first trial, and the range was narrowed to 111–131 in order to increase the signal to noise ratio of the target species. The voltage was set to 3.5 kV for positive ESI. The capillary temperature was 300 °C.

MS/MS experiments were performed for the Orgueil extract using a hybrid quadrupole-Orbitrap mass spectrometer (Q-Exactive Plus, Thermo Fisher Scientific) with HPLC and ionization conditions identical to those used for the full-scan analyses. The extracted positive ion $m/z$ (for example, for uracil, 113.03 ± 0.2) was reacted with high-energy collision $N_2$ gas to produce fragmented ions, and the mass range of $m/z$ 50–160 was monitored using an Orbitrap MS with a mass resolution of ~140,000. The MS/MS measurements were not performed for the Ryugu samples due to the unavailability of the sample solution.

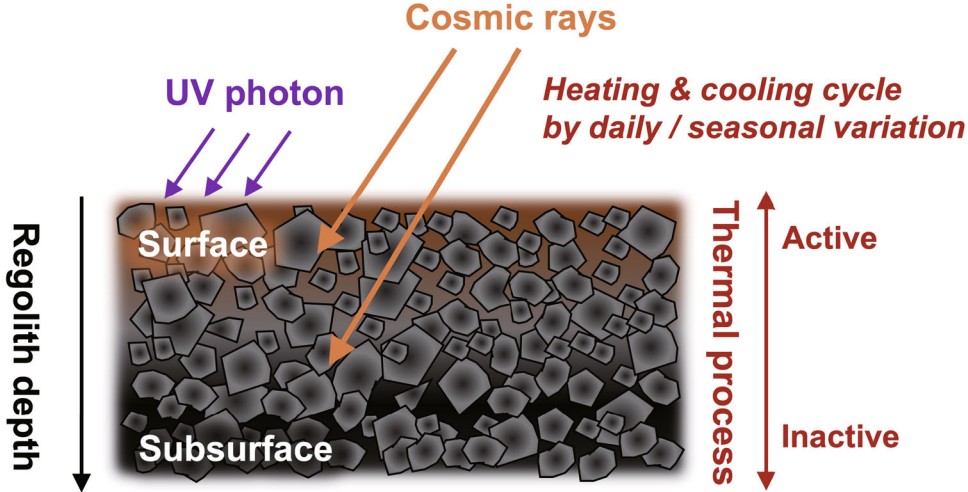

**Fig. 8 | Conceptual view of surface and sub-surface physico-chemical features along with regolith depth.** Conceptual diagram of the effects of thermal cycling in the diurnal (i.e., variation in daytime and nighttime for max. and min. temperatures; -100 °C)[4] and seasonal illumination cycles, which are constant exposure processes by photon and cosmic ray irradiation on the surface and subsurface regolith at Ryugu. The purple and orange arrows indicate ultraviolet and cosmic rays, respectively, and the length of each line is a qualitative model of the penetration depth into the regolith. Kitazato et al.[19] estimated that the current seasonal thermal skin depth of Ryugu is up to -1.1 metre by the model simulation, whereas the total accumulated heat storage within the daily thermal skin depth is estimated to be ~ few centimeter scale below the surface layer[57].

The methanol extract from the Ryugu samples was introduced into a nano liquid chromatograph (UltiMate 3000 RSLCnano, Thermo Fisher Scientific) equipped with a reversed phase C18 column (Aurora, 250 mm length × 75 mm i.d., particle size 1.6 μm, IonOptics) and a nanoelectrospray ion source. The eluent program for the nanoLC was as follows: solvent A (water + 0.1% formic acid) and solvent B (acetonitrile + 0.1% formic acid) = 99:1 at $t$ = 0–1 min, followed by a linear gradient of A:B = 75:25 at $t$ = 25, and the additional linear gradient of A:B = 15:85 at $t$ = 30, maintained at this ratio for 10 min. The flow rate was 400 nL/min. The column temperature was 40 °C. The mass spectra were recorded in positive ESI mode with an $m/z$ range of 60.5–600 and a spray voltage of 1.6 kV. The capillary temperature was 250 °C.

For cross-validation of the detailed analysis of nucleobases in the Ryugu hot water extracts (unhydrolyzed fraction), we conducted the capillary electrophoresis-high resolution mass spectrometry (CE-HRMS) using an ω Scan package method (Human Metabolome Technologies (HMT), Inc., Japan) described previously[55]. Briefly, CE-HRMS analysis was carried out using an Agilent 7100 CE capillary electrophoresis system (Agilent Technologies, Inc., Santa Clara, CA, USA) equipped with a Q Exactive Plus (Thermo Fisher Scientific Inc., Waltham, MA, USA), Agilent 1260 isocratic HPLC pump, Agilent G1603A CE-MS adapter kit, and Agilent G1607A CE-ESI-MS sprayer kit (Agilent Technologies). The systems were controlled by Agilent MassHunter workstation software LC/MS data acquisition for 6200 series TOF/ 6500 series Q-TOF version B.08.00 (Agilent Technologies) and Xcalibur (Thermo Fisher Scientific), and connected by a fused silica capillary (80 cm total length × 50 μm i.d.) with the electrophoresis buffer (H3301-1001, HMT) as the electrolyte. The spectrometer was scanned from $m/z$ 60 to 900 in positive mode[55]. Peaks were extracted using MasterHands, automatic integration software (Keio University, Tsuruoka, Yamagata, Japan) in order to obtain peak information including $m/z$, peak area, and migration time (MT)[56].

## Data availability
The raw data are available at the following databases in the Hayabusa2 Science Data Archives (DARTS, https://www.darts.isas.jaxa.jp/planet/ project/hayabusa2/). We declare that all these database publications are compliant with ISAS data policies (www.isas.jaxa.jp/en/ researchers/data-policy/).

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

## Acknowledgements

The Hayabusa2 project has been led by JAXA (Japan Aerospace
Exploration Agency) in collaboration with DLR (German Space Center)
and CNES (French Space Center) and supported by NASA (National
Aeronautics and Space Administration) and ASA (Australian Space
Agency). We also thank the project members of the Extraterrestrial
Samples Curation Center (ESCuC) and the Astromaterials Science
Research Group (ASRG) at Institute of Space and Astronautical Science
(ISAS) for processing the sample. This research is partly supported by
the Japan Society for the Promotion of Science (JSPS) under KAKENHI
grant numbers; 21H04501 & 21H05414 (Y.O.), 21J00504 (T.K.), 21KK0062
(Y.T.), 20H00202 (H.N.). J.P.D. and D.P.G. thank NASA for support of the
Consortium for Hayabusa2 Analysis of Organic Solubles. This study was
partly conducted by the official collaboration agreement through the
joint research project with JAMSTEC, Keio University and HMT Inc. This
study was conducted in accordance with the Joint Research Promotion
Project at the Institute of Low Temperature Science, Hokkaido University
(21G008 and 22G008 to Y.O., Y.T., and H.N.).

## Author contributions

Y.O., T.K., Y.Ta, and H.N. outlined the working flow in this study. H.N. and
Y.Ta conducted sequential extraction procedure, distributed the SOM
samples. Y.O., T.K., and H.N. conducted the analysis of the Orgueil
meteorite from the Natural History Museum of Denmark. Y.O., T.K., and
H.N. performed the molecular analysis by the LC/Orbitrap system. H.N.,
N.O.O., N.O., and Y.Ta conducted the analysis of elemental and isotopic
compositions. N.O.O. and N.O. provided the series of authentic C, N
isotope standards. H.N. provided the series of H, O isotope standards.
K.Sas, H.S., and Y.Ta conducted the analysis by the CE/Orbitrap system.
D.P.G. and J.P.D. authorized the importance of nitrogen-containing
heterocyclic compounds. H.N., Y.Ta, and J.P.D. designed the imple-
mentation of the SOM scheme and the initial analysis timelines (~31-May-
2022). S.Tac, H.Yu, H.N., T.Na, T.No, R.O., H.Ya, and K.Sak managed the
initial analysis processes and wrote the paper. M.A., T.Y., M.N., K.Y., A.N.,
A.M., T.O., and T.U. curated samples. M.Y., T.S., S.Tan, F.T., S.N., S.W.,
and Y.Ts contributed to the sample collection at Ryugu. The Hayabusa2-
initial-analysis SOM team members are shown in the Supplementary
Information. All authors discussed the results and commented on the
manuscript.

## Competing interests

The authors declare no competing interests.

## Additional information

**Supplementary information** The online version contains
supplementary material available at

Yasuhiro Oba or Yoshinori Takano.

**Peer review information** *Nature Communications* thanks Christian
Potiszil and the other, anonymous, reviewer(s) for their contribution to
the peer review of this work. Peer reviewer reports are available.

[1]Institute of Low Temperature Science (ILTS), Hokkaido University, N19W8, Kita-ku, Sapporo 060-0819, Japan. [2]Biogeochemistry Research Center (BGC),
Japan Agency for Marine-Earth Science and Technology (JAMSTEC), Natsushima, Yokosuka 237-0061, Japan. [3]Institute for Advanced Biosciences (IAB),
Keio University, Kakuganji, Tsuruoka, Yamagata 997-0052, Japan. [4]Human Metabolome Technologies Inc., Kakuganji, Tsuruoka, Yamagata 997-0052,
Japan. [5]Solar System Exploration Division, NASA Goddard Space Flight Center, Greenbelt, MD 20771, USA. [6]Department of Earth and Planetary Sciences,
Kyushu University, 744 Motooka, Nishi-ku, Fukuoka 819-0395, Japan. [7]UTokyo Organization for Planetary and Space Science (UTOPS), University of Tokyo,
7-3-1 Hongo, Tokyo 113-0033, Japan. [8]Institute of Space and Astronautical Science (ISAS), Japan Aerospace Exploration Agency (JAXA), Sagamihara 252-
5210, Japan. [9]Department of Earth and Planetary Sciences, Hokkaido University, Sapporo 060-0810, Japan. [10]Department of Earth Material Science,
Tohoku University, Sendai 980-8578, Japan. [11]Department of Earth and Planetary Sciences, Kyoto University, Kyoto 606-8502, Japan. [12]Department of
Earth and Planetary Systems Science, Hiroshima University, 739-8526 Higashi-Hiroshima, Japan. [13]Kanagawa Institute of Technology, Atsugi 243-0292,
Japan. [14]Graduate School of Environmental Studies, Nagoya University, Nagoya 464-8601, Japan. [15]A list of authors and their affiliations appears in the
Supplementary Information. ✉e-mail: oba@lowtem.hokudai.ac.jp; takano@jamstec.go.jp

