## [Peer Review File · Nature Communications]

Uracil in the carbonaceous asteroid (162173) RyuguReviewer #1 (Remarks to the Author):

The manuscript by Yasuhiro Oba and co-authors reports on the results of their HPLC-HRMS study of Ryugu sample extracts. The study indicates the presence of uracil, an important nucleobase used by all life on Earth for genetic functions. Other nitrogen heterocyclic compounds, some of which are also of biological significance, are also reported. The findings from two groups of samples, each from a different touchdown site, are then put into the context of their sampling sites. One sample is from a touchdown at the very surface of Ryugu, while the other is from nearby an artificial impact crater and should thus have some subsurface material. The authors conclude that the differences in nucleobase concentration between the sample groups relates to their differential exposure to UV/cosmic rays while on the Ryugu asteroid. The potential formation pathways concerning uracil and the other N-heterocycles are also discussed.

Overall, the paper is of high significance to the astrobiology/cosmochemistry/planetary science community and reports on the highly anticipated Ryugu samples. The conclusions are interesting and contribute to our understanding of the processes affecting organic matter on primitive asteroids. I believe that the data can adequately support the conclusions of the paper and therefore suggest that it is accepted, as long as a number of concerns (detailed below) are adequately addressed.

Major Points

1). While the references are mostly appropriate, I think the following should be included alongside reference 8 in the summary paragraph, because there are already published results concerning amino acids from Ryugu: Nakamura et al, 2022, which can be found at: https://www.jstage.jst.go.jp/article/pjab/98/6/98_PJA9806B-01/_article/-char/ja/.

The same ref should also appear on line 64, because it was an extensive geochemical analysis of the Ryugu samples and also showed through many different analytical techniques applied to 16 particles that Ryugu is similar to Orgueil and thus CI chondrites.

The ref should again appear on line 68 as the study also reported the presence of N-containing compounds in Ryugu, which were likely N-heterocycles.

Also, the ref should appear on line 185, as the study also showed differences between touch down site 1 and 2 samples and came to a similar conclusion about the role of cosmic rays, which were backed up by Ne isotopic data.

And also ref the study on line 217 where you talk about urea, this was identified in this study as well.

2). While the samples are briefly described in Figure 1A, some more in-text details about the samples should be given. Is there any background information reported elsewhere about the representativeness of the samples or any other data reported by yourselves or the Phase 1 curation team that can be briefly described or at least mentioned, so readers can go have a look? Are there any spectroscopic or SEM data for these particles?

I am interested in hearing about how the samples were separated into separate aggregates? Was care taken to ensure a representative sample was used for each extraction procedure, by for example ensuring a mix of different particle sizes and particles with different surface characteristics were included in both extraction procedures?

3). Was any acid required to extract the Nucleobases? I know Callahan et al., 2011 used 95% formic acid, but in Oba et al., 2022 and the Naraoka et al., (submitted to Science) there is no mention of acid. However, in Furukawa et al., 2019 2% HCl is used to extract sugars and I think one of the fractions is used in Oba et al., 2022. It's just that I thought

some acid or base was necessary to extract nucleobases from meteorites as I didn't think they were very soluble in pure water. Am I wrong in thinking this or was some acid used in the extraction?

I'm also still a little confused, could you detect other nucleobases (than uracil) here for Orgueil? Orgueil was found to have guanine, adenine and purine by Callahan and Oba et al., 2022 was able to detect similar amounts of nucleobases to Callahan for the Murchison extract 1. If you didn't detect other nucleobases, is this partly because Murchison extract 1 in Oba et al., 2022 was extracted with 2% HCl (like in Furukawa et al., 2019) and extract 2, which had much less nucleobases was not extracted with acid? Sorry if I have got confused here and please clarify it for me.

Also, in Oba et al., 2022, it is mentioned, when comparing to Callahan et al., 2011, that "If such severe extraction conditions are more suitable for the extraction of nucleobases from meteorites, it is likely that the actual concentrations of nucleobases in our meteorite samples are higher than those reported here." Did you try to investigate whether your method underestimates the abundances of nucleobases in meteorites and if so, did you try to quantify this and apply a correction to the Ryugu results?

Minor Points

1). It would be good to mention about the difference in uracil concentration between TD1 and TD2 samples and the role of UV/cosmic rays in the summary. I think this is an important conclusion of the study.

2). Line 123: While I don't have a problem with estimating an upper limit for cytosine, I do think the wording could be changed to emphasize here that this is an estimate based on an estimate. Maybe try something like:

The upper limit for the cytosine concentration can be estimated from the difference in the concentration of uracil before and after acid hydrolysis (estimated from Orgueil), to be 4 and 11 ppb in the A0106 and C0107 samples, respectively.

3). Line 205: Other studies have also suggested that Ryugu may be composed of comet-like materials or even formed from a comet, e.g. Nakamura et al., 2022, Potiszil et al., 2020 and Mura et al., 2022. It might be good to reference at least one of them here and probably several.

4). Please include the data for your Orgueil sample in Fig. 2, in order to allow an easy comparison.

In their review of the first version of this manuscript, reviewer #1 added some comments to the manuscript file. These comments were forwarded to the authors, who replied as included in this Peer Review File.

Reviewer #2 (Remarks to the Author):

The present manuscript reports on the analysis of returned Ryugu samples for organic compounds of astrobiological interest. It has the potential to be a really important manuscript but there are a few issues that need to be addressed.

First, there are a number of spelling and grammar errors that should be corrected as they detract from the readability of the paper.

Second, some of the data are presented in a confusing manner. The authors should double check that all of their data really show what they want to be showing. A couple examples are given below to illustrate this.

Extended data figure 4 is confusing to me. From the legend for B, it looks as though this is a MS/MS experiment. Were the authors not able to run these as SRM or MRM experiments? The presence of so many peaks in the lower m/z chromatograms makes the data less conclusive, as there are apparently several isobaric compounds present in the samples such that the peaks do not necessarily have to come from the parent "uracil" molecule.

Extended data figure 8 is also confusing. Which of these peaks do the authors think are the actual analytes of interest? From top to bottom there are way too many peaks in the 5-10 minute range for them all to be closely-related structural isomers at a given molecular weight. Particularly for the low m/z compounds like m/z = 113.0346. And a similar envelope of stuff shows up in that 5-10 minute window for all m/z traces. Once you get to the C9 – C15 section there is a fairly large dispersed envelope of "peaks" that slowly shifts to later retention times but if they really are alkylated uracil analogs I would expect many more discrete peaks rather than just smears of signals. The shading of the figure, too, implies that every peak/signal in the chromatogram is real but the data are not convincing that any of these signals are anything other than noise. What is the mass range that these plots show (1 ppm or some other range)?

Finally, the title does not really fit the substance of the paper. Uracil is one of the compounds that was reported here, but the presence of many heterocyclic compounds appears to be the main point of the paper. The authors should consider renaming the article to better fit the data that they report here.

Reviewer #3 (Remarks to the Author):

Oba et al. present a manuscript on the detection of uracil in samples from the carbonaceous asteroid Ryugu. This is the first analysis showing its presence in the samples returned by the Hayabusa 2 mission. This result is important for the origin of this body. They also compare this result to a CI meteorite showing its presence in Orgueil. They also study the distribution of alkyl uracil in extracts of two different samples from Ryugu.

Despite the clear scientific interest of this work, major clarifications have to be done before acceptance for publication.

Major:

1. The samples from Orgueil were treated with water or by acid hydrolysis. The authors show that the treatment changes the amount of uracil, which implies a possible alteration during acid hydrolysis. Why was the Ryugu sample analyzed only after acid hydrolysis? This treatment clearly damaged the original sample, which prevents a coherent discussion of Ryugu chemistry. The authors need to justify why they did not analyze their Ryugu sample after only water extraction because of the results they observed on Orgueil. Add to the main text what is written in the method section.
2. In the same trend, alkylated uracil was analyzed after methanol extraction. Why was uracil not searched for in this extract and compared to hot water and acidic sample treatment?
3. MS/MS was performed on Orgueil samples to verify that fragment patterns are coherent between Orgueil and uracil standard. This strengthens the presence of uracil in Orgueil. However, for Ryugu samples, this MS/MS was not performed, and the only justification of uracil presence is the fact that MS/MS was performed on Orgueil and that a peak presenting the same retention time with the same stoichiometric formulae is present in the Ryugu sample. Why don't the authors apply the same strategy on Ryugu. It is necessary to justify it, because a significant doubt taints this detection without this

MS/MS.

4. Authors discuss about the possible degradation of cytosine in uracil during acidic hydrolysis that could explain the variation of uracil quantity observed between water extract and acidic hydrolysis. From that, they extrapolate an upper limit for the amount of cytosine in Ryugu samples. This is purely speculative. At least, authors should verify the amount of cytosine in Orgueil by changing their analysis protocol and verify the hypothesis that a fraction of the uracil increase observed is related to the degradation of cytosine.

5. Authors explain that low temperature chemistry may have led to uracil detected in Ryugu and Orgueil. As explained by authors, Ryugu can have been heated up to 100°C, which it is not low temperature, and the sample treatment at 105°C + acidic-hydrolysis clearly modify the organic matrix as they demonstrate on Orgueil. It seems therefore difficult to discuss the origin of these compounds since finally the processing of the samples blurs the native presence of uracil in the samples. Isotopic measurement on specific molecules could give information on low temperature formation if uracil is enriched in heavy isotopes.

6. On ICA, no MS/MS are present. As demonstrated in ruf et al. 2019 DOI : 10.3847/2041-8213/ab59df, high resolution mass spectrometry with LC is not enough to unambiguously demonstrate the presence of a compound since a high number of isomers are present in such samples ruf et et al. 2019 DOI : 10.3390/life9020035. Picolinamide was confirmed by MS:MS. Where are the data?

7. Regarding the alkylated uracil homologues, where is the evidence that they are uracil based, and not just isomers. Without MS/MS and only on HRMS it is difficult to conclude that they are homologue series of uracil. On the bar diagrams, what are the uncertainties on the intensities? Are they statically significant? The signals are close to the noise.

8. As a general comment. It is unfortunate that a strong treatment was applied to the Ryugu samples, because as the authors themselves say, this treatment probably altered the original sample, making the mechanistic discussion highly speculative. For example, the authors take the example of HMT. It is well known that if HMT is placed under the conditions of the sample treatment, HMT can be partially degraded and lead to molecular diversity with molecules resembling that detected here. Vinogradoff et al. DOI: 10.1016/j.icarus.2017.12.019.

9. Furthermore, a lack of sample cannot justify the remaining uncertainties about the actual presence of uracil in the Ryugu samples. If it is not possible to strengthen this detection, this work cannot be published in Nature Communication.

Minor:

P2 line 56, change "prebiotic building locks" by "molecules of prebiotic interest"

P4 line 132, add to ref 11 ref DOI : 10.3847/2041-8213/ab59df

P5 line 140, there is no "prebiotic evolution" on asteroids. Rewrite like "prebiotic evolution on the early Earth.

P6, line 185 add ref De Marcellus et al. DOI : 10.1093/mnras/stw2292 and Danger et al. 2022 DOI: 10.1051/0004-6361/202244191

Replies to Reviewer's comments on the manuscript #NCOMMS-22-37350-T

We deeply appreciate the constructive review comments from three Reviewers on our manuscript (NCOMMS-22-37350-T) entitled "Uracil in the carbonaceous asteroid (16293) Ryugu". Firstly, we have chosen the option of "Transparent peer review system" to post the open discussions during the entire review processes (*Nature Communications*, doi: 10.1038/ncomms10277). Secondly, we carefully read all of the comments in the reviews and modified the original version of the manuscript based on their helpful feedback. The changes we made based on the Reviewer's comments are noted in **red font** in the revised manuscript/supporting information. Our point-by-point responses to each comment (in Times New Roman font) and denoted below following the reviewer's comments (in Arial font). In the revision, "Extended Data Figure" was rephrased as "Supplementary Figure" which is more appropriate for Nature Communications.

Reviewer #1 (Remarks to the Author):

The manuscript by Yasuhiro Oba and co-authors reports on the results of their HPLC-HRMS study of Ryugu sample extracts. The study indicates the presence of uracil, an important nucleobase used by all life on Earth for genetic functions. Other nitrogen heterocyclic compounds, some of which are also of biological significance, are also reported. The findings from two groups of samples, each from a different touchdown site, are then put into the context of their sampling sites. One sample is from a touchdown at the very surface of Ryugu, while the other is from nearby an artificial impact crater and should thus have some subsurface material. The authors conclude that the differences in nucleobase concentration between the sample groups relates to their differential exposure to UV/cosmic rays while on the Ryugu asteroid. The potential formation pathways concerning uracil and the other N-heterocycles are also discussed.

Overall, the paper is of high significance to the astrobiology/cosmochemistry/planetary science community and reports on the highly anticipated Ryugu samples. The conclusions are interesting and contribute to our understanding of the processes affecting organic matter on primitive asteroids. I believe that the data can adequately support the conclusions of the paper and therefore suggest that it is accepted, as long as a number of concerns (detailed below) are adequately addressed.

Our reply: Thank you very much for your positive comments on our manuscript.

Major Points

1). While the references are mostly appropriate, I think the following should be included alongside reference 8 in the summary paragraph, because there are already published results concerning amino acids from Ryugu: Nakamura et al, 2022, which can be found at: https://www.istage.ist.go.jp/article/pjab/98/6/98_PJA9806B-01/article/-char/ja/.

The same ref should also appear on line 64, because it was an extensive geochemical analysis of the Ryugu samples and also showed through many different analytical techniques applied to 16 particles that Ryugu is similar to Orgueil and thus CI chondrites.

The ref should again appear on line 68 as the study also reported the presence of N-containing compounds in Ryugu, which were likely N-heterocycles. Also, the ref should appear on line 185, as the study also showed differences between touch down site 1 and 2 samples and came to a similar conclusion about the role of cosmic rays, which were backed up by Ne isotopic data. And also ref the study on line 217 where you talk about urea, this was identified in this study as well.

Our reply: We agree with those comments. Given the remarkable progress that has been made on the analysis of the Ryugu samples, we have updated the discussion and the new knowledge to reflect them, including the recent original paper (i.e., Nakamura et al., 2022 as ref. 9). Please confirm the contents of the line numbers (57, 72, 76, 198, 232-233).

[Ref.#9] Nakamura, E., Kobayashi, K., Tanaka, R., Kunihiro, T., Kitagawa, H., Potiszil, C., Ota, T., Sakaguchi, C., Yamanaka, M., Ratnayake, D.M., Tripathi, H., Kumar, R., Avramescu, M.-L., Tsuchida, H., Yachi, Y., Miura, H., Abe, M., Fukai, R., Furuya, S., Hatakeda, K., Hayashi, T., Hitomi, Y., Kumagai, K., Miyazaki, A., Nakato, A., Nishimura, M., Okada, T., Soejima, H., Sugita, S., Suzuki, A., Usui, T., Yada, T., Yamamoto, D., Yogata, K., Yoshitake, M., Arakawa, M., Fujii, A., Hayakawa, M., Hirata, N., Hirata, N., Honda, R., Honda, C., Hosoda, S., Iijima, Y.-i., Ikeda, H., Ishiguro, M., Ishihara, Y., Iwata, T., Kawahara, K., Kikuchi, S., Kitazato, K., Matsumoto, K., Matsuoka, M., Michikami, T., Mimasu, Y., Miura, A., Morota, T., Nakazawa, S., Namiki, N., Noda, H., Noguchi, R., Ogawa, N., Ogawa, K., Okamoto, C., Ono, G., Ozaki, M., Saiki, T., Sakatani, N., Sawada, H., Senshu, H., Shimaki, Y., Shirai, K., Takei, Y., Takeuchi, H., Tanaka, S., Tatsumi, E., Terui, F., Tsukizaki, R.,

Wada, K., Yamada, M., Yamada, T., Yamamoto, Y., Yano, H., Yokota, Y., Yoshihara, K., Yoshikawa, M., Yoshikawa, K., Fujimoto, M., Watanabe, S.-i. and Tsuda, Y. (2022) On the origin and evolution of the asteroid Ryugu: A comprehensive geochemical perspective. *Proceedings of the Japan Academy, Series B* 98, 227-282.

Refs. 8 & 31 were also updated on this manuscript.

2). While the samples are briefly described in Figure 1A, some more in-text details about the samples should be given. Is there any background information reported elsewhere about the representativeness of the samples or any other data reported by yourselves or the Phase 1 curation team that can be briefly described or at least mentioned, so readers can go have a look? Are there any spectroscopic or SEM data for these particles?

Our reply: We have added a new Fig. 1-C and additional description of the spectroscopic data for the Ryugu sample as follows (lines 100-104), “**The FTIR spectral analysis of those Ryugu samples indicated that the signals derived from hydrous silicate minerals (i.e., 2.72 μm for OH absorption, Fig. 1-C) and organic matter (i.e., 3.1 μm and 3.4 μm for CH and NH absorption bands, respectively) were mostly homogeneous between A0106 and C0107 and had similar characteristics to a profile of CI chondritic features⁷.**”

We also declared that we restored the raw data on the properties of the asteroid Ryugu from the following databases in the Hayabusa2 Science Data Archives (DARTS, <https://www.darts.isas.jaxa.jp/planet/project/hayabusa2/>).

I am interested in hearing about how the samples were separated into separate aggregates? Was care taken to ensure a representative sample was used for each extraction procedure, by for example ensuring a mix of different particle sizes and particles with different surface characteristics were included in both extraction procedures?

Our reply: Given the average composition of the samples (Fig. 1-A), which are fine grain powders, we took care to ensure that a representative sample was used in each extraction procedure. In this context, average nondestructive spectra are additionally shown in Fig. 1-C as we mentioned above.

3). Was any acid required to extract the Nucleobases? I know Callahan et al., 2011 used

95% formic acid, but in Oba et al., 2022 and the Naraoka et al., (submitted to Science) there is no mention of acid. However, in Furukawa et al., 2019 2% HCl is used to extract sugars and I think one of the fractions is used in Oba et al., 2022. It's just that I thought some acid or base was necessary to extract nucleobases from meteorites as I didn't think they were very soluble in pure water. Am I wrong in thinking this or was some acid used in the extraction?

Our reply: In Oba et al. (2022), indeed we used 2% HCl for the extraction of nucleobases from the Murchison #1 sample, while we did not use any acid to extract nucleobases from the Murchison #2 sample. From both Murchison samples, nucleobases have been extracted successfully (see Oba et al. 2022, Tables 1 and 2), but the molecular distribution was different between the Murchison #1 and #2 samples. This could be due to the different extraction method used. However, it is also possible that the differences in the distribution of nucleobases was due to sample heterogeneity since the Murchison meteorite specimens extracted were from different fragments.. Note that even if nucleobases were extracted with 2% HCl, they were dissolved in pure H₂O just before the analysis using HPLC/HRMS.

I'm also still a little confused, could you detect other nucleobases (than uracil) here for Orgueil? Orgueil was found to have guanine, adenine and purine by Callahan and Oba et al., 2022 was able to detect similar amounts of nucleobases to Callahan for the Murchison extract 1. If you didn't detect other nucleobases, is this partly because Murchison extract 1 in Oba et al., 2022 was extracted with 2% HCl (like in Furukawa et al., 2019) and extract 2, which had much less nucleobases was not extracted with acid? Sorry if I have got confused here and please clarify it for me.

Our reply: No, we did not detect any other nucleobases in the Orgueil extract from this study. As the reviewer mentioned, other nucleobases were detected in previous studies. This discrepancy would be mainly caused by the very small amount of Orgueil meteorite (~10 mg) extracted in the present study. We expect other nucleobases could be easily detected in the aqueous extract from Orgueil if we extracted 1 gram or more of the meteorite, as was demonstrated in Oba et al. (2022).

In Oba et al., 2022, it is mentioned, when comparing to Callahan et al., 2011, that "If such severe extraction conditions are more suitable for the extraction of nucleobases from meteorites, it is likely that the actual concentrations of nucleobases in our meteorite

samples are higher than those reported here." Did you try to investigate whether your method underestimates the abundances of nucleobases in meteorites and if so, did you try to quantify this and apply a correction to the Ryugu results?

Our reply: Thank you very much for the careful reading of our recent publication. We did not investigate the extraction efficiency of nucleobases from Ryugu given the limited mass of sample available for this study. We plan to investigate the nucleobase extraction efficiencies using different methods on carbonaceous meteorites in the future, although it is not clear if those results could be extrapolated to the Ryugu samples to apply an accurate correction since they are different materials.

Minor Points

1). It would be good to mention about the difference in uracil concentration between TD1 and TD2 samples and the role of UV/cosmic rays in the summary. I think this is an important conclusion of the study.

Our reply: We added the following sentence to the summary paragraph. **“The observed difference in the concentration of uracil between A0106 and C0107 may be related to the possible differences in the degree of alteration induced by energetic particles such as ultraviolet photons and cosmic rays.”**

2). Line 123: While I don't have a problem with estimating an upper limit for cytosine, I do think the wording could be changed to emphasize here that this is an estimate based on an estimate. Maybe try something like:

The upper limit for the cytosine concentration can be estimated from the difference in the concentration of uracil before and after acid hydrolysis (estimated from Orgueil), to be 4 and 11 ppb in the A0106 and C0107 samples, respectively.

Our reply: Modified as suggested.

3). Line 205: Other studies have also suggested that Ryugu may be composed of comet-like materials or even formed from a comet, e.g. Nakamura et al., 2022, Potiszil et al., 2020 and Mura et al., 2022. It might be good to reference at least one of them here and probably several.

Our reply: We added Nakamura et al. (2022) (ref. #9) here (line 220).

4). Please include the data for your Orgueil sample in Fig. 2, in order to allow an easy comparison.

Our reply: We added the Orgueil data to Figure 2 as suggested. Accordingly, Extended Data Figures 3 and 5 were removed.

Reviewer #2 (Remarks to the Author):

The present manuscript reports on the analysis of returned Ryugu samples for organic compounds of astrobiological interest. It has the potential to be a really important manuscript but there are a few issues that need to be addressed.

Our reply: Thank you very much for your positive comments.

First, there are a number of spelling and grammar errors that should be corrected as they detract from the readability of the paper.

Our reply: We fixed all spelling and grammatical errors throughout the manuscript.

Second, some of the data are presented in a confusing manner. The authors should double check that all of their data really show what they want to be showing. A couple examples are given below to illustrate this.

Extended data figure 4 is confusing to me. From the legend for B, it looks as though this is a MS/MS experiment. Were the authors not able to run these as SRM or MRM experiments? The presence of so many peaks in the lower m/z chromatograms makes the data less conclusive, as there are apparently several isobaric compounds present in the samples such that the peaks do not necessarily have to come from the parent "uracil" molecule.

Our reply: We did a MS/MS experiment, and the MS/MS data were shown in Extended Data (Supplementary) Figure 3 in the revision. We were not able to run SRM or MRM experiments due to the limited sample amount. Since the concentration of uracil in the Orgueil extract was not very high, and the fragmentation of uracil should lead to lower intensity daughter fragments relative to the parent mass of uracil itself, the peak height for the fragment is inevitably small. Nevertheless, the observed peak height for the uracil fragment was clearly higher than the background noise and since no interfering peaks were observed for imidazole carboxylic acids or other structural isomers of uracil, we concluded that the observed peak at a retention time of ~16 min was only derived from uracil.

Extended data figure 8 is also confusing. Which of these peaks do the authors think are the actual analytes of interest? From top to bottom there are way too many peaks in the

5-10 minute range for them all to be closely-related structural isomers at a given molecular weight. Particularly for the low m/z compounds like $m/z = 113.0346$. And a similar envelope of stuff shows up in that 5-10 minute window for all m/z traces. Once you get to the C9 – C15 section there is a fairly large dispersed envelope of “peaks” that slowly shifts to later retention times but if they really are alkylated uracil analogs I would expect many more discrete peaks rather than just smears of signals. The shading of the figure, too, implies that every peak/signal in the chromatogram is real but the data are not convincing that any of these signals are anything other than noise. What is the mass range that these plots show (1 ppm or some other range)?

Our reply: Indeed, we are not sure how much the observed peaks are derived from alkylated uracil analogues. Rather, alkylated analogues of imidazole carboxylic acids and other structural isomers may also be present; however, we cannot assign each peak to any specific molecules since we do not have any standard reagents, which are required for the peak assignments to a molecule, for such highly alkylated analogues of uracil and imidazole carboxylic acid. Nevertheless, the observed trend (i.e. a gradual shift with increasing carbon numbers) suggests that the observed peaks belong to similar group(s). Since the observed peak heights at each m/z value are clearly higher than those obtained in the blank experiments, we are confident that these are not background noise but derived from molecules in the extract. To avoid confusion by readers, we changed the legend for Extended Data Figure 8 (Supplementary Fig. 6 in the revision) as follows: **“Detection of $C_nH_{2n-4}N_2O_2$ molecules from the extract of A0106. Mass chromatograms at the m/z corresponding to $C_nH_{2n-4}N_2O_2$ molecules with the carbon number (n) of 4 to 15. Alkylated analogues of uracil and imidazole-carboxylic acids are represented by this molecular formula. Numbers on the left indicate the relative intensities for each chromatogram”**. The mass range of these plots was within 5 ppm.

Finally, the title does not really fit the substance of the paper. Uracil is one of the compounds that was reported here, but the presence of many heterocyclic compounds appears to be the main point of the paper. The authors should consider renaming the article to better fit the data that they report here.

Our reply: Indeed, we prefer the present title that emphasizes the detection of the nucleobase uracil in the Ryugu sample since this discovery has high astrobiological significance and will be of interest to a broader audience. However, as the reviewer pointed out, we detected not only uracil but also other N-heterocycles. We would like

to have an opinion from the Editor on the title of our manuscript by suggesting the title, “Uracil and other N-heterocycles in the carbonaceous asteroid (162173) Ryugu”.

Reviewer #3 (Remarks to the Author):

Oba et al. present a manuscript on the detection of uracil in samples from the carbonaceous asteroid Ryugu. This is the first analysis showing its presence in the samples returned by the Hayabusa 2 mission. This result is important for the origin of this body. They also compare this result to a CI meteorite showing its presence in Orgueil. They also study the distribution of alkyl uracil in extracts of two different samples from Ryugu. Despite the clear scientific interest of this work, major clarifications have to be done before acceptance for publication.

Our reply: Thank you very much for the constructive comments. We have included additional evidence from the cross-validation of the detection of uracil in the Ryugu extract using different analytical approaches. We will respond to the comments one by one below.

Major:

1. The samples from Orgueil were treated with water or by acid hydrolysis. The authors show that the treatment changes the amount of uracil, which implies a possible alteration during acid hydrolysis. Why was the Ryugu sample analyzed only after acid hydrolysis? This treatment clearly damaged the original sample, which prevents a coherent discussion of Ryugu chemistry. The authors need to justify why they did not analyze their Ryugu sample after only water extraction because of the results they observed on Orgueil. Add to the main text what is written in the method section.

Our reply: Since the search for amino acids was indeed one of the main targets for the soluble organic matter analysis team in the Hayabusa2 project, almost all of the hot water extract was subjected to acid hydrolysis to release bound amino acids, as has been performed previously for the extraction of amino acids from carbonaceous meteorites (e.g. Glavin et al. 2021). After the successful detection of amino acids in the acid hydrolysate (Naraoka et al. in press), the remaining acid hydrolyzed water extract (a couple of tens of microliters) was used for the analysis of N-heterocycles, including nucleobases. We did want to analyze nucleobases in the hot water extracts from the Ryugu samples before acid hydrolysis. However, the limited sample mass did not allow us to do so, unfortunately. Instead, we tried to analyze nucleobases in the hot water extracts from Orgueil both before and after acid hydrolysis to strengthen our conclusion that uracil should be present in the acid hydrolyzed, hot water extracts from the Ryugu

samples. The use of Orgueil as a reference material is reasonable based on the fact that Ryugu is very similar to CI chondrites in terms of elemental composition (Yokoyama et al. 2022). We added the information on the sample amount (~10 mg) to the main text (L96).

2. In the same trend, alkylated uracil where analyzed after methanol extraction. Why uracil was not searched for in this extract and compared to hot water and acidic sample treatment?

Our reply: The methanol extract was analyzed using a nanoLC/orbitrapHRMS. In general, although nanoLC has a very good potential to detect tiny amounts of molecules in a sample, its potential to distinguish structural isomers is not high enough, as has been shown in the mass chromatograms where the peak separation was not good (Supplementary Figure 6 in the revision). We actually searched for it and several peaks were identified as shown in Supplementary Figure 6 (in the revision). However, in the present case, not only uracil but also 4-imidazole-carboxylic acid and 2-imidazole-carboxylic acid appear around the same retention time on the mass chromatogram at $m/z = 113.0346$. Hence, although several peaks are observed in the mass chromatogram, we were not able to assign the observed peaks to any specific molecules.

3. MS/MS was performed on Orgueil samples to verify that fragment pattern are coherent between Orgueil and uracil standard. This strengthens the presence of uracil in Orgueil. However, for Ryugu samples, this MS/MS was not performed, and the only justification of uracil presence is the fact that MS/MS was performed on Orgueil and that a peak presenting the same retention time with the same stoichiometric formulae is present in Ryugu sample. Why don't the authors apply the same strategy on Ryugu. It is necessary to justify it, because a significant doubt taints this detection without this MS/MS.

Our reply: As mentioned above, the sample amount of Ryugu for this study was very limited. Hence, we were not able to perform the MS/MS analysis for the Ryugu extract. Moreover, even if we could make MS/MS measurements for the Ryugu samples, we suspect any reliable data would not be obtained due to the very low concentration of uracil. For Orgueil, the hot water extract and its acid hydrolysate was used for the analysis of N-heterocycles only. While for Ryugu, the extract was mainly used for amino acid detection as mentioned above. Nevertheless, the fact that a peak was observed at the same retention time (~16 min) in the Ryugu extract with that of Orgueil

at $m/z = 113.0346$ measured under the identical conditions, strongly supports our conclusion that the observed peak is uracil. We have analyzed aqueous extracts from several carbonaceous meteorites (Oba et al. 2022) under the same analytical conditions and have confirmed that no interfering mass peaks appear at the same retention time as uracil, which strengthens the case for the absence of other structural isomers of uracil at the same retention time in the Ryugu extracts. I understand the reviewer's statement that MS/MS measurements are ideal to firmly assign a specific molecule in the sample; however, even without MS/MS results, under the conditions where the number of possible structural isomers is limited, the observed peak can be reasonably assigned to a specific molecule by comparison of the mass peak retention time with the standard chemical reagent.

4. Authors discuss about the possible degradation of cytosine in uracil during acidic hydrolysis that could explain the variation of uracil quantity observed between water extract and acidic hydrolysis. From that, they extrapolate an upper limit for the amount of cytosine in Ryugu samples. This is purely speculative. At least, authors should verify the amount of cytosine in Orgueil by changing their analysis protocol and verify the hypothesis that a fraction of the uracil increase observed is related to the degradation of cytosine.

Our reply: Of course, to derive an upper limit is generally based on some assumptions. However, since this is based on the experimental observations that cytosine can be converted to uracil upon extensive hydrothermal and/or acidic treatments, the use of "upper limit" here in this context is reasonable. The purpose of this procedure is NOT to exactly estimate how much cytosine was converted to uracil after acid hydrolysis, BUT just to mention one of the possibilities for the origin of uracil. To prevent misleading by readers, we rephrased the sentences as follows (L138-141): "The upper limit for the concentrations of uracil precursors including cytosine can be estimated from... respectively. There could be other formation mechanisms of uracil in the Ryugu sample other than the hydrolysis of its chemical precursors".

5. Authors explain that low temperature chemistry may have led to uracil detected in Ryugu and Orgueil. As explained by authors, Ryugu can have been heated up to 100°C, which it is not low temperature, and the sample treatment at 105°C + acidic-hydrolysis clearly modify the organic matrix as they demonstrate on Orgueil. It seems therefore difficult to discuss the origin of these compounds since finally the processing of the

samples blurs the native presence of uracil in the samples. Isotopic measurement on specific molecules could give information on low temperature formation if uracil is enriched in heavy isotopes.

Our reply: We just mentioned here that uracil may be partly synthesized by photochemical reactions in interstellar ices as has been demonstrated in our previous study (Oba et al. 2019, Nat. Comm.). Isotopic measurements could be helpful to constrain the origin of uracil in the Ryugu extract, but it was not possible due to the low concentrations of extractable uracil and sample mass required to make the compound specific isotopic measurement (e.g. ~ 1 μg of uracil is required for the analysis of both C and N isotopic compositions (Koga et al. unpublished). Given that the concentration of uracil in C0107 is 32 ppb, we require more than 30 g of Ryugu samples, which are far above the total abundance of the recovered Ryugu samples ~ 5.4 g). We expect the detected uracil has multiple sources for its origin, such as low-T photochemistry and hydrothermal activities on the parent body and during extraction. We just proposed here one of such possibilities.

With regards to the temperature variation and regolith depth profiles, we updated the skin depth issues in Supplementary Figure 8 with the new reference of Shimaki et al. (2020).

[Ref. 57] Shimaki, Y., Senshu, H., Sakatani, N., Okada, T., Fukuhara, T., Tanaka, S., Taguchi, M., Arai, T., Demura, H., Ogawa, Y., Suko, K., Sekiguchi, T., Kouyama, T., Hasegawa, S., Takita, J., Matsunaga, T., Imamura, T., Wada, T., Kitazato, K., Hirata, N., Hirata, N., Noguchi, R., Sugita, S., Kikuchi, S., Yamaguchi, T., Ogawa, N., Ono, G., Mimasu, Y., Yoshikawa, K., Takahashi, T., Takei, Y., Fujii, A., Takeuchi, H., Yamamoto, Y., Yamada, M., Shirai, K., Iijima, Y.-i., Ogawa, K., Nakazawa, S., Terui, F., Saiki, T., Yoshikawa, M., Tsuda, Y. and Watanabe, S.-i. (2020) Thermophysical properties of the surface of asteroid 162173 Ryugu: Infrared observations and thermal inertia mapping. *Icarus* 348, 113835.

6. On ICA, no MS/MS are present. As demonstrated in ruf et al. 2019 DOI : 10.3847/2041-8213/ab59df, high resolution mass spectrometry with LC is not enough to unambiguously demonstrate the presence of a compound since a high number of isomers are present in such samples ruf et et al. 2019 DOI : 10.3390/life9020035. Picolinamide was confirmed by MS:MS. Where are the data?

Our reply: We realize that with increasing molecular weight, the number of structural isomers with specific molecular formulae also increases. However, as has been demonstrated in our previous study (Oba et al. 2022), the number of structural isomers with $C_4H_4N_2O_2$ was not very high in carbonaceous meteorites previously analyzed; in the case of Murchison, for example, only three species were positively identified at $m/z = 113.0346$, which corresponds to the protonated ion of $C_4H_4N_2O_2$. These assignments were confirmed by MS/MS measurements (Oba et al. 2022). This is also true for Orgueil where only three peaks at this m/z value were observed (Supplementary Figure 3 in the revision), and whose chemical composition is similar to Ryugu (Yokoyama et al. 2022). These results strongly suggest that uracil, 2-imidazolecarboxylic acid and 4-imidazolecarboxylic acid are the three major species with this m/z value in carbonaceous meteorites and Ryugu. Thus, our analyses of carbonaceous meteorites suggest that the observed three peaks at $m/z = 113.0346$ in the Ryugu extract can be reasonably assigned to two imidazole carboxylic acids and uracil even without supporting MS/MS measurements.

As for the MS/MS of picolinamide, please see the following figure:

We do not think this figure is necessary in the manuscript.

7. Regarding the alkylated uracil homologues, where is the evidence that they are uracil based, and not just isomers. Without MS/MS and only on HRMS it is difficult to conclude that they are homologue series of uracil. On the bar diagrams, what are the uncertainties on the intensities? Are they statically significant? The signals are close to the noise.

Our reply: As the reviewer mentioned, it is not possible to conclude that all of the detected peaks are derived from alkylated uracil homologues. So, to prevent misleading the reader, we modified the caption in Extended Data Figure 8 (Supplementary 6 in the revised version) as follows: “**Detection of $C_nH_{2n-4}N_2O_2$ molecules from the extract of A0106. Mass chromatograms at the m/z corresponding to $C_nH_{2n-4}N_2O_2$ molecules with the carbon number (n) of 4 to 15. Alkylated analogues of uracil and imidazole-carboxylic acids are represented by this molecular formula. Numbers on the left indicate the relative intensities for each chromatogram**”.

Accordingly, Figure 3 and Extended Data Figure 9 (Supplementary Fig. 7 in the revised version) were also modified to precisely represent the obtained data. In general, the peak intensity in the sample was much higher than those observed in the blank experiments. Hence, we are confident that most of the detected peaks are indigenous to the Ryugu samples. As for the uncertainty of the peak intensities, the nanoLC analysis generally has good reproducibility. Although each Ryugu sample was analyzed only once due to the limited sample availability for the measurement, the similar distribution in each class of N-heterocycles between A0106 and C0107 (Supplementary Figure 7 in the revised version) strongly implies that the analysis is highly reproducible and hence it is statistically meaningful.

8. As a general comment. It is unfortunate that a strong treatment was applied to the Ryugu samples, because as the authors themselves say, this treatment probably altered the original sample, making the mechanistic discussion highly speculative. For example, the authors take the example of HMT. It is well known that if HMT is placed under the conditions of the sample treatment, HMT can be partially degraded and lead to molecular diversity with molecules resembling that detected here. Vinogradoff et al. DOI: 10.1016/j.icarus.2017.12.019.

Our reply: We agree with the reviewer. However, for the analysis of amino acids in meteorites, returned samples, and also the organic residues generated from laboratory analog experiments, such acid hydrolysis treatment has been applied to the samples, and the detected amino acids identified in these experiments have always been identified as such, even if they were formed from other chemical precursors during the extraction process.. In that sense, the detected uracil in the acid hydrolysate from the Ryugu hot water extract can be considered as “uracil” in the Ryugu samples. To further rationalize the present results, we added the crucial data for the presence of uracil in the hot water

extract before acid hydrolysis, as described in our next reply shown below.

9. Furthermore, a lack of sample cannot justify the remaining uncertainties about the actual presence of uracil in the Ryugu samples. If it is not possible to strengthen this detection, this work cannot be published in Nature Communication.

Our reply: We recognize the importance of cross-validation to assure the analytical accuracy of the present data. For further confirmation of evidence of indigenous uracil, we have analyzed the hot water extracts (unhydrolyzed fraction) from the Ryugu samples (A0106 & C0107) by capillary electrophoresis (CE) coupled with orbitrap-type ultra-high-resolution mass spectrometry. Firstly, we have succeeded in detecting the signal of uracil based on the migration time of CE separation and the exact mass identification for both Ryugu hot water extracts (A0106 & C0107). The detected signal of uracil was clearly above the background level, meaning that uracil was present in the hot water extract prior to acid hydrolysis. Secondly, the observed peak area for uracil in the A0106 hot water extract was two times smaller than the uracil peak in the C0107 extract, which is consistent with the trend for the relative concentration of uracil in the acid hydrolysates of the hot water extracts from both samples (Table 1). Thus, we have emphasized the new result and discussion in the main text as follows (L117-120), “**The simultaneous detection of uracil from the A0106 and C0107 extracts by a different separation technique using a capillary electrophoresis-high resolution mass spectrometry (CE-HRMS) instrument is important evidence that reliably supports the analytical detection of uracil with HPLC/ESI-HRMS as discussed above (Fig. 2C).**”

Note: The unhydrolyzed fraction was not measured by LC/HRMS due to the limited sample volume.

Method description in L 305-319.

For cross-validation of the detailed analysis of nucleobases in the Ryugu hot water extracts (unhydrolyzed fraction), we conducted the capillary electrophoresis-high resolution mass spectrometry (CE-HRMS) using an ω Scan package method (Human Metabolome Technologies (HMT), Inc., Japan) described previously⁵⁵. Briefly, CE-HRMS analysis was carried out using an Agilent 7100 CE capillary electrophoresis system (Agilent Technologies, Inc., Santa Clara, CA, USA) equipped with a Q Exactive Plus (Thermo Fisher Scientific Inc., Waltham, MA, USA), Agilent 1260 isocratic HPLC pump, Agilent G1603A CE-MS adapter kit, and Agilent G1607A CE-ESI-MS sprayer kit (Agilent Technologies). The systems were controlled by Agilent MassHunter

workstation software LC/MS data acquisition for 6200 series TOF/6500 series Q-TOF version B.08.00 (Agilent Technologies) and Xcalibur (Thermo Fisher Scientific), and connected by a fused silica capillary (50 μm i.d. \times 80 cm total length) with the electrophoresis buffer (H3301-1001, HMT) as the electrolyte. The spectrometer was scanned from m/z 60 to 900 in positive mode, respectively⁵⁵. Peaks were extracted using MasterHands, automatic integration software (Keio University, Tsuruoka, Yamagata, Japan) in order to obtain peak information including m/z , peak area, and migration time (MT)⁵⁶.

[Ref. 55] Sasaki, K., Sagawa, H., Suzuki, M., Yamamoto, H., Tomita, M., Soga, T. and Ohashi, Y. (2019) Metabolomics platform with capillary electrophoresis coupled with high-resolution mass spectrometry for plasma analysis. *Analytical Chemistry* 91, 1295-1301.

[Ref. 56] Sugimoto, M., Wong, D.T., Hirayama, A., Soga, T. and Tomita, M. (2010) Capillary electrophoresis mass spectrometry-based saliva metabolomics identified oral, breast and pancreatic cancer-specific profiles. *Metabolomics* 6, 78-95.

Then, the caption for the new Fig. 2C is as follows, “(C) Cross-validation and high-resolution molecular identification of uracil obtained from hot water extracts (unhydrolyzed fraction) for Ryugu A0106 and C0107 by using capillary electrophoresis (CE) coupled with high resolution orbitrap mass spectrometry (CE-HRMS). The signal of red color represents uracil on the migration time (18.4 min).”

Minor:

P2 line 56, change “prebiotic building locks” by “molecules of prebiotic interest”

Our reply: Modified as suggested.

P4 line 132, add to ref 11 ref DOI : 10.3847/2041-8213/ab59df

Our reply: We appreciate the constructive comment, we cited it as requested.

[Ref.21] Ruf, A., Lange, J., Eddhif, B., Geffroy, C., d’Hendecourt, L.L.S., Poinot, P. and Danger, G. (2019) The challenging detection of nucleobases from pre-accretionary astrophysical ice analogs. *The Astrophysical Journal Letters* 887, L31.

P5 line 140, there is no “prebiotic evolution” on asteroids. Rewrite like “prebiotic evolution on the early Earth.

Our reply: We rewrite as “chemical evolution” on the line.

We note that as long as chemical reactions take place abiologically, it should be also called as “prebiotic” evolution.

P6, line 185 add ref De Marcellus et al. DOI : 10.1093/mnras/stw2292 and Danger et al. 2022 DOI: 10.1051/0004-6361/202244191

Our reply: Cited as requested in the context.

[Ref. 33] de Marcellus, P., Fresneau, A., Brunetto, R., Danger, G., Duvernay, F., Meinert, C., Meierhenrich, U.J., Borondics, F., Chiavassa, T. and Le Sergeant d'Hendecourt, L. (2017) Photo and thermochemical evolution of astrophysical ice analogues as a source for soluble and insoluble organic materials in Solar system minor bodies. *Monthly Notices of the Royal Astronomical Society* 464, 114-120.

[Ref. 34] Danger, G., Ruf, A., Javelle, T., Maillard, J., Vinogradoff, V., Afonso, C., Schmitz-Afonso, I., Remusat, L., Gabelica, Z. and Schmitt-Kopplin, P. (2022) The transition from soluble to insoluble organic matter in interstellar ice analogs and meteorites. *Astronomy & Astrophysics*. **667**, A120.

Reviewer #2 (Remarks to the Author):

In my opinion, the authors have satisfactorily addressed the comments of me and the other reviewers.

Reviewer #3 (Remarks to the Author):

Dear Author,

First of all, I would like to thank you for the time you have taken to respond to the various comments.

My main concern was the clarification of the identification of the different compounds observed from the analyses.

I very much appreciate the use of CE-HRMS which confirms the presence of uracil and gives new clues to its presence, even though no MS/MS was possible due to lack of sample.

I therefore recommend the publication in Nature communication.